## [Editor Report · Decision Letter 0]

2 Sep 2025

Dear Dr Glebov,

Thank you for submitting your revised manuscript entitled "Rapid long-range synaptic remodelling in hyperacute ischemic stroke" for consideration as a Research Article by PLOS Biology. I have had a chance to discuss your revised study and your response to reviewers with the Academic Editor and I am writing to let you know that we would like to send your submission back to the original reviewers for another look.

Once your full submission is complete, your paper will undergo a series of checks in preparation for peer review. After your manuscript has passed the checks it will be sent out for review. To provide the metadata for your submission, please Login to Editorial Manager (https://www.editorialmanager.com/pbiology) within two working days, i.e. by Sep 04 2025 11:59PM.

Kind regards,

Luke

Lucas Smith, Ph.D.

Senior Editor

PLOS Biology

lsmith@plos.org

---

## [Decision Letter · Decision Letter 1]

17 Oct 2025

Dear Dr Glebov,

Thank you for your patience while we considered your revised manuscript "Rapid long-range synaptic remodelling in hyperacute ischemic stroke" for consideration as a Research Article at PLOS Biology. Your revised study has now been evaluated by the PLOS Biology editors, the Academic Editor and the original reviewers.

As you will see in the comments below, both reviewers 1 and 2 are fully satisfied with the revision. Reviewer 3 also indicates that the paper has been strengthened, but s/he has a few lingering concerns which will need to be carefully addressed. Of note, reviewer 3 has commented that you should clarify how body temperature was maintained during MK801 administration and if body temperature was not carefully controlled during stroke and MK801 administration, this experiment needs to be included. Overall we agree with that assessment.

In light of the reviews, which you will find at the end of this email, we are pleased to offer you the opportunity to address the remaining points from the reviewer 3 in a revision that we anticipate should not take you very long. However, if you need to generate new data to address his/her comments, please do let us know and we are happy to provide more time. We will then assess your revised manuscript and your response to the reviewers' comments with our Academic Editor aiming to avoid further rounds of peer-review, although we might need to consult with the reviewers, depending on the nature of the revisions.

**IMPORTANT - SUBMITTING YOUR REVISION**

*Resubmission Checklist*

*Published Peer Review*

*PLOS Data Policy*

*Blot and Gel Data Policy*

Sincerely,

Luke

Lucas Smith, Ph.D.

Senior Editor

PLOS Biology

lsmith@plos.org

REVIEWS:

Reviewer #1, Christoph Harms (note, reviewer 1 has signed this review): The authors have increased the transparency of their study. I have no further concerns.

Reviewer #2: The authors succesfully replied to my methodological concerns.

Reviewer #3: Throughout the manuscript the long-distance synaptic changes are referred to as 'enhanced'. This is not appropriate language and not supported by the data as presented. Enhancement has a positive valence associated with it, but none of the changes as reported in the paper can be demonstrated to be enhanced, just changed. A more suitable descriptor should be chosen by the authors because the alteration of synaptic genes, changes in expression of receptors and increased synaptic vesicle cycling could actually be detrimental to brain function and stroke. for example, increased expression of K channels would likely be inhibitory for neuronal function, which would not be considered enhancement.

during the administration of MK801, was the body temperature of the animal carefully controlled? This needs to be described in the methods and is critical because known effects of MK801 include hyperthermia, which can cause broad changes in brain function and has been reported to alter gene expression. Importantly, if body temperature was not carefully controlled during stroke and MK801 administration, this experiment needs to be included.

Fig 1A does not seem to be representative of the data. There is an almost complete loss of BSN labelling (for example) at 2 hrs in OGD, but this is not seen in Fig B

---

## [Editor Report · Decision Letter 2]

24 Nov 2025

Dear Dr Glebov,

Thank you for your patience while we considered your revised manuscript "Rapid long-range synaptic remodelling in hyperacute ischemic stroke" for publication as a Research Article at PLOS Biology. Your revised study has been evaluated by the PLOS Biology editors and the Academic Editor.

We understand from your responses to reviewer 3, that while you controlled body temperature during the MCAO experiments with a heating pad, that you have not formally measured or controlled the body temperature during the MK801 experiments, as requested in the last round of review. We understand that you instead cite a body of literature showing that MK801's effects on body temperature can be heterogeneous and that they are generally minor. Unfortunately, we do not think that this response has gone far enough in addressing reviewer 3's comment. We would therefore like to invite you to complete one last revision, to address this point with additional data.

After discussing this point with the Academic Editor and with my colleagues on the editorial team, we would not require that you repeat the characterization of all outcomes after MK801 administration. However, at minimum, we think you will need to repeat the MK801 study as previously performed, and measure whether, in your hands and in this context, the drug affects body temperature, or not. Depending on the results of those studies, we think you should then add additional discussion about whether this is a likely confound affecting the results of your experiments, or not.

As an additional note, I actually could not find the details of the MK801 administration studies in your methods section. Please do take a moment to update the methods to include all relevant details about how that experiment was performed, including dosing, method of delivery, timing, etc. Indeed, please double check that the methods section includes the relevant details for all of the experiments conducted in this study.

Given that we think that additional experimental work is needed to address Reviewer 3's concerns, we are offering 3 months for you to complete this next revision. Please note, that given the extent of revision needed, we cannot make a decision about publication until we have seen the revised manuscript and your response to the reviewers' comments. This will be the last round of experimental revisions that we will allow.

Please email us (plosbiology@plos.org) if you have any questions or concerns, or would like to request an extension.

**IMPORTANT - SUBMITTING YOUR REVISION**

*Re-submission Checklist*

*Published Peer Review*

*PLOS Data Policy*

*Blot and Gel Data Policy*

Sincerely,

Luke

Lucas Smith, Ph.D.

Senior Editor

PLOS Biology

lsmith@plos.org

---

## [Editor Report · Decision Letter 3]

17 Dec 2025

Dear Dr Glebov,

Thank you for your patience while we considered your revised manuscript "Rapid long-range synaptic remodelling in hyperacute ischemic stroke" for publication as a Research Article at PLOS Biology. This revised version of your manuscript has been evaluated by the PLOS Biology editors and the Academic Editor, who is satisfied by the changes made in the most recent revision.

Based on our Academic Editor's assessment of your revision we are likely to accept this manuscript for publication. However, before we can do so, we need you to address a number of data and other policy-related requests in a last short revision. These are detailed below.

IMPORTANT - Please address the following editorial requests:

1) TITLE: We would like to suggest a tweak to your title, to add a verb and a bit more detail about the study. If you agree, we suggest you change your title to:

'Ischemic stroke triggers brain-wide synaptic remodeling within four hours'

2) ABSTRACT: Please note that per journal policy, the model system/species studied should be clearly stated in the abstract of your manuscript.

3) ETHICS STATEMENT: Please update your ethics statement, in your methods section, to include the protocol number of the animal care and use protocol, approved by the Qingdao University IACUC.

4) DATA: Thank you for providing your mass spec and RNA seq data on proteomeXchange and GEO. For the RNA-seq data, please provide me with a reviewer token so I can access this data and make sure it meets our requirements.

5) DATA: Thanks also for providing the other underlying data for your study as a supplemental table. Please add a sentence to every relevant figure legend referencing this file. We also ask that you update your data availability statement to reference this data.

For example, you could change the Data Availability statement to read:

Mass spectrometry proteomics data is available from the ProteomeXchange (https://proteomecentral.proteomexchange.org), dataset identifier PXD058834.

RNAseq data is avialable from GEO (https://www.ncbi.nlm.nih.gov/geo/), accession number GSE283465.

All other relevant data are contained within the manuscript and its supplemental materials.

Other raw data is available from the corresponding Author upon reasonable request

6) CODE: Per journal policy, if you have generated any custom code during the course of this investigation, please make it available without restrictions. Please ensure that the code is sufficiently well documented and reusable, and that your Data Statement in the Editorial Manager submission system accurately describes where your code can be found.

We expect to receive your revised manuscript within three weeks.

*Published Peer Review History*

*Press*

Sincerely,

Luke

Lucas Smith, Ph.D.

Senior Editor

lsmith@plos.org

PLOS Biology

---

## [Editor Report · Decision Letter 4]

6 Jan 2026

Dear Dr Glebov,

Thank you for the submission of your revised Research Article "Ischemic stroke triggers brain-wide synaptic remodeling within four hours" for publication in PLOS Biology and thank you for addressing our last editorial requests in this revision. On behalf of my colleagues and the Academic Editor, Richard Daneman, I am pleased to say that we can in principle accept your manuscript for publication, provided you address any remaining formatting and reporting issues. These will be detailed in an email you should receive within 2-3 business days from our colleagues in the journal operations team; no action is required from you until then. Please note that we will not be able to formally accept your manuscript and schedule it for publication until you have completed any requested changes.

**PLEASE NOTE: I saw that you provided an updated data and code availability statement in your manuscript. I have taken the liberty of moving this to the relevant section of our editorial manager system, as that is the version that will be published with your study. I removed the bit about Oe Biotech's analysis pipeline, as that is better left in the methods section, rather than the data availability statement. The current version reads:

Custom code for gene expression analysis has been uploaded to Zenodo (https://zenodo.org/records/17987265). Mass spectrometry proteomics data is available at ProteomeXchange (https://proteomecentral.proteomexchange.org, dataset identifier PXD058834). RNAseq data is available at GEO (https://www.ncbi.nlm.nih.gov/geo/, accession number GSE283465). Numerical data is presented in Table S1. Other raw data is available from the Corresponding Author upon reasonable request.

^Please do take a moment to double check that everything looks good, in our online system, after this change.

PRESS

Sincerely,

Lucas Smith, Ph.D.

Senior Editor

PLOS Biology

lsmith@plos.org